# Characterization of the Toxigenic Potential of *Bacillus cereus sensu lato* Isolated from Raw Berries and Their Products

**DOI:** 10.3390/foods12214021

**Published:** 2023-11-03

**Authors:** Márcia Oliveira, Marta Carvalho, Paula Teixeira

**Affiliations:** 1Department of Food Hygiene and Technology, University of León, 24071 León, Spain; msouo@unileon.es; 2Laboratório Associado, CBQF—Centro de Biotecnologia e Química Fina, Escola Superior de Biotecnologia, Universidade Católica Portuguesa, Rua Diogo Botelho 1327, 4169-005 Porto, Portugal; marta_carvalho1992@hotmail.com

**Keywords:** *Bacillus cereus*, foodborne spore-formers, enterotoxins, emetic toxins, toxigenic profile, cytotoxicity

## Abstract

*Bacillus cereus* is estimated to be responsible for 1.4–12% of all food poisoning outbreaks worldwide. The objective of this study was to investigate the toxigenic potential of 181 isolates of *B. cereus* previously recovered from different types of berries and berry products (strawberries, raspberries, blackberries, and blueberries) by assessing the presence of enterotoxin genes (*hblA*, *hblC*, *hblD*, *nheA*, *nheB*, *nheC*, and *cytK*) and an emetic toxin cereulide synthetase gene (*ces*). The cytotoxic activity on Caco-2 cells was also evaluated for the two isolates containing the gene *cytK*. Twenty-three toxigenic profiles were found. The *nheABC* (91.7%) and *hblACD* (89.0%) complexes were the most prevalent among the isolates, while the *cytK* and *ces* genes were detected in low percentages, 1.1% and 3.3%, respectively. In addition, the *nheABC*/*hblACD* complex and *ces* genes were detected in isolates recovered throughout the production process of blackberries and strawberries. The cytotoxic activity on Caco-2 cells was also observed to be greater than 60% for isolates containing the *cytK* gene.

## 1. Introduction

Increased fruit and vegetable consumption is driven by factors like the growing awareness of their health benefits. Consequently, there has been a rise in foodborne illness cases and outbreaks linked to their consumption [1]. Fresh and frozen berries have attracted particular attention as a potential source of foodborne illness [2]. Although most incidents are associated with viruses and parasites, bacterial outbreaks have also been documented [3].

The *Bacillus cereus* group, also called *B. cereus sensu lato* (*s.l.*), is a group of genetically closely related species that includes pathogenic and non-pathogenic species [4]. As far as the authors know at the time of writing, the group comprises nine closely related bacterial species including, *B. anthracis* [5], *B. cereus* sensu stricto (*s.s.*), *B. cytotoxicus* [6], *B. mycoides* [7], *B. pseudomycoides* [8], *B. thuringiensis*, *B. toyonensis* [9], *B. weihenstephanensis* [7], and *B. wiedmannii* [10]. Their genetic similarity has been widely studied and is characterized by having significant implications for human health, agriculture, and food processing [11]. The main species recognized as human pathogens are *B. cereus s.s.*, most commonly associated with food poisoning, *B. anthracis*, the causative agent of anthrax, and *B. cytotoxicus*, a thermotolerant species also associated with food poisoning [12].

In a shift towards a strain-based approach to characterization and risk assessment, a new taxonomic framework for *B. cereus s.l.* was proposed by Carroll et al. [13]. The aim was to provide a more accurate assessment of the pathogenic potential of *B. cereus s.l.* isolates while maintaining genomic species definitions. Within this updated framework, the previously known *B. cereus s.l.* species were reclassified into eight genomic species. In addition, three distinct biovars were created: Emeticus, Anthracis, and Thuringiensis, based on the presence of genes associated with specific characteristics. More recently, Zhang et al. [14] conducted a study that examined the species diversity of *B. cereus* strains and their phylogenetic relationships concerning virulence factors. The aim of the research was to systematically evaluate the distribution of virulence genes and analyze the structures and functions of virulence proteins. By assessing enterotoxicity through *nheABC*, *hblACD*, and *cytK* gene expression, the study classified the strains into three levels of toxicity. The results suggested that *B. cereus* strains have evolved into clusters characterized as non-toxic risk and toxic risk, further subdivided into medium-high-risk and medium-low-risk subclusters.

Strains of *B. cereus* are widespread in the environment and can be found in soil, raw fruits and vegetables, raw herbs, and dry and processed foods [15,16,17]. They are spore-forming bacteria and some strains are psychrotrophic and extremely resistant to various environmental and processing conditions, resulting in the contamination of various foods after processing [18,19].

*Bacillus cereus* gastrointestinal diseases are caused by a large number of virulence factors that are not yet fully understood. The level of expression of genes that play a role in its pathogenicity (e.g., genes encoding for toxins, such as hemolysins, enterotoxins, and emetic toxins, as well as enzymes involved in tissue degradation) determine the actual risk of food poisoning [4]. Two types of food poisoning are considered, either by ingestion of large numbers of bacterial cells and/or spores from contaminated food (diarrhoeal type) or by ingestion of food contaminated with a pre-formed toxin (emetic type) [20,21,22]. Diarrhoea is caused by enterotoxins released in the small intestine by cells that survive gastric passage or during vegetative growth after spore germination [23]; when pre-formed in foods, these enterotoxins most likely do not contribute to the disease, as they are considered sensitive towards heat, acids or proteases [24]. Diarrhoea is generally attributed to different enterotoxins, including non-hemolytic enterotoxin NHE [25], hemolysin BL (HBL) [25], and cytotoxin K (CytK) [26]. Emetic syndrome is associated with the ingestion of the dodecadepsipeptide cereulide, which is pre-formed in food before ingestion (as reviewed by Yang et al. [27]). Symptoms (vomiting and nausea) typically appear 30 min to 6 h after eating contaminated food. This symptomatology resembles that of staphylococcal poisoning [20]. Because of its resistance to heat, acids, and proteolysis, cereulide remains active even after exposure to heat treatment in contaminated food samples or the digestive environment of the stomach [27,28,29]. According to Park et al. [30] emetic *B. cereus* strains exposed to temperatures higher than their optimum growth temperature could potentially develop tolerance to disinfectants and become more virulent.

The true incidence of *B. cereus* food poisoning may be underestimated due to the misdiagnosis of the disease, which is symptomatically similar to other types of food poisoning, and the generally short duration of illness [31]. Still, according to data from 2021, *B. cereus* was reported as the fifth most common causative agent of foodborne outbreaks in the EU [2].

The level of *B. cereus* in fruits and fruit products in generally low [15]. It is generally understood that food products below 10^4^ to 10^5^ cells or spores per gram are not considered to pose a significant risk [32,33]. However, when used as ingredients, these products can potentially contaminate various foods, such as starchy meals, creating favourable conditions for *B. cereus* growth and toxin production.

Little information is available on the toxigenic profile of *B. cereus* present in fruits, particularly in raw berries and their products. To this regard, the objective of this work was to determine the presence of enterotoxin (*hblA*, *hblC*, *hblD*, *nheA*, *nheB*, *nheC*, and *cytK*) and the emetic toxin (*ces*) genes in 181 *B. cereus* isolates collected from berry samples, from raw berries to processed fruits.

## 2. Materials and Methods

### 2.1. Bacterial Isolates

The *B. cereus s.l.* isolates used in this study, hereafter referred to as *B. cereus*, were collected from the work of Oliveira et al. [15]. A total of 181 isolates (Table 1) (bacilli displaying typical growth for the *B. cereus* group, i.e., rough and dry colonies with violet pink background surrounding an egg yolk precipitation, and hemolysis on blood agar) were isolated from berry fruit samples (strawberries (n = 25), raspberries (n = 74), blackberries (n = 43) and blueberries (n = 39)) in three fruit flavour production steps (RM: raw material; IP: intermediate product; FP: final product). *B. cereus* NVH 0075-95 (*nhe* reference), *B. cytotoxicus* NVH 391-98 (*cytK* reference) [34], *B. cereus* DSMZ 4313 (*hbl/nhe* reference) [35], and *B. cereus* DSMZ 4312 (cereulide and *nhe* reference) [36] were used as reference strains.

### 2.2. DNA Extraction

All *B. cereus* isolates were grown by streaking on Tryptone Soy Agar (TSA, Biokar Diagnostics, Allonne, France) plates and incubating at 30 °C for 24 h. A single colony was then inoculated into Tryptone Soy Broth (TSB, Biokar Diagnostics) and incubated at 30 °C for 18 h. Subsequently, DNA of each isolate was extracted using the GRS Genomic DNA Kit (Grisp, Porto, Portugal) according to the manufacturer’s protocol for Gram-positive bacteria. The PCR screening was carried out to detect the presence of seven enterotoxigenic genes (*hblA*, *hblC*, *hblD*, *nheA*, *nheB*, *nheC*, and *cytK*), and one emetic gene (*ces*).

### 2.3. Primers and Multiplex PCR Reaction

Table 2 shows the sequences of the primer pairs used in this study to amplify the virulence factor genes of *B. cereus*. All primers used in this study have previously been used in multiplex PCR assays for the simultaneous detection of enterotoxigenic and emetic genes [37,38,39,40].

Multiplex PCR was performed in an automatic thermal cycler (Bio-Rad, Hercules, CA, USA) under the following optimized cycling program (Table 3): an initial denaturation step of 3 min at 95 °C; 35 cycles of denaturation at 94 °C for 30 s; annealing at 58 °C for 45 s; an extension at 72 °C for 1.5 min; and a final extension at 72 °C for 5 min. For each reaction tube, 25 ng of DNA was used. Ultrapure water was used for all negative control reactions and for the preparation of the PCR mixture. All mixtures for the amplification of sequences encoding toxins contained 2 μL of template DNA (25 ng), 10× Taq Buffer + KCl (ThermoFisher Scientific, Waltham, MA, USA), 10 mM of each deoxynucleoside triphosphate (dNTPs) (Bioron, Römerberg, Germany), 25 mM of MgCl2 (ThermoFisher Scientific), 50 μM of each primer, and 1 U μL^−1^ of Taq DNA polymerase (ThermoFisher Scientific). Amplified fragments, GRS Ladder 100 bp (Grisp), and positive and negative controls were analyzed using electrophoresis on 1.5% agarose in 1× buffer (108 g Trisbase L^−1^, 55 g boric acid L^−1^ and 40 mL of 0.5 M EDTA, pH 8.0) at 80 V over 1.5 h.

### 2.4. Cytotoxic Activity

The cytotoxic activity of bacterial supernatants on the human colon adenocarcinoma cell line Caco-2 (American Type Culture Collection ECACC 86010202) was evaluated using the method described by Gdoura-Ben Amor et al. [41] with some modifications. Caco-2 cells were cultivated on 96-well microplates at 37 °C under 5% CO_2_ atmosphere for 3 days in Eagle’s minimal essential medium (Lonza, Verviers, Belgium) supplemented with 1% (*v*/*v*) of non-essential amino acids (Biosera, Boussens, France), 1% (*v*/*v*) of pyruvate (Lonza), and 20% (*v*/*v*) of fetal bovine serum (FBS; Biowest, Nuaillé, France). Of the 181 *B. cereus* isolates, only isolates that had the *cytK* gene were tested (*B. cereus* isolate profile II and XIX) and two different controls were used: negative control (BactoFlavor^®^ScarLet, Chr. Hansen A/S, Hoersholm, Denmark) and positive control (*B. cytotoxicus* NVH 391-98). These bacteria were grown for 18 h and 5 days at 30 °C in Brain Heart Infusion (BHI, Biokar Diagnostics, France) with additional 6 g L^−1^ of Yeast Extract (YE, Biokar, France), without agitation. Cell suspensions were centrifuged (10 min, 7000 rpm, 4 °C) and the supernatants were filtered through 0.22 μm sterile filter units (Sartorius, Göttingen, Germany). After removal of the culture medium, Caco-2 cells were washed 3 times with phosphate buffered saline (PBS, 0.1 M, pH 7.4; Sigma, Darmstadt, Germany), incubated over 3 h with 50 μL of each bacterial filtrate and then rinsed with PBS and fixed with 2% (*v*/*v* in PBS) formaldehyde (Sigma-Aldrich, Darmstadt, Germany) at 4 °C for 30 min. Formaldehyde was removed and the remaining cells were stained for 20 min at room temperature with 80 μL of 2% crystal violet solution (*v*/*v* in PBS) (Merck, Darmstadt, Germany). Cells were rinsed three times with distilled water, the crystal violet solution was released from the cells by adding 200 μL of 50% (*v*/*v*) ethanol in water and shaking the microplates at room temperature for 45 min and, lastly, were transferred into new microplates and the optical density (OD) was measured at 630 nm. The cytotoxic activity was expressed as a percentage of inhibition compared with the negative control, calculated as follows:Cytotoxic activity (%)=(OD control−OD isolate)/(OD control)×100

*B. cereus* isolates were considered cytotoxic when the percentage of inhibition was higher than 50%. Tests and controls were carried out in triplicate on the same microplate.

### 2.5. Statistical Analysis

Data analyses and plotting were carried out in R environment (http://www.r-project.org accessed on 15 June 2022). A barplot representing the prevalence of toxin profiles in each sample type was drawn with the function ‘geom_bar’ from the ‘ggplot2’ R package.

Data analysis for cytotoxic activity of *B. cereus* strains on Caco-2 cells was performed using IBM SPSS software (version 28.0). Statistical differences were analyzed for significance by one-way ANOVA using Tukey’s multiple range tests. Statistical significance was determined at the *p* < 0.05 levels.

## 3. Results and Discussion

### 3.1. Distribution of Virulence Genes among B. cereus Isolates

The virulence potential of *B. cereus* recovered from berries (strawberries, raspberries, blackberries, and blueberries) and berries products [15] was investigated by screening the presence of seven diarrhoeal toxin-encoding genes (*hblACD*, *nheABC*, *cytK*) and one emetic toxin-encoding gene (*ces*), as shown in Figure 1. It should be highlighted that, in some isolates, not all genes are always detected using PCR due to the presence of polymorphisms in the sequences of genes of the HBL and NHE complexes [38]. Remarkably, all 181 *B. cereus* isolates analysed in this study showed at least one of the toxin genes examined.

Twenty-three different toxin profiles were found among the *B. cereus* isolates and their prevalence and distribution are shown in Table 4 and Table 5 and in Figure 2. Consistent reports indicate the presence of multiple toxin profiles [14,42,43,44]; however, the most prevalent profile varies between studies. Enterotoxin genes were found more frequently in blackberry, raspberry, and strawberry samples than in blueberries. The complex *nheABC*/*hblACD* (toxin profile VI), causing diarrhoeal-type disease, showed the highest prevalence (44.2%; 80/181), followed by the toxin profile V (*nheAB*/*hblACD*) with a prevalence of 8.3% (Table 4); moreover, at least one gene of each complex was detected in a percentage of 91.7% (166/181) and 89.0% (161/181) for NHE and HBL, respectively (Table 5). These enterotoxin genes were detected almost always above 50%, with *hblD*, *nheA*, and *nheB* showing the highest prevalence (85.6, 85.1, and 81.8%, respectively) regardless of the fruit product. Additionally, genes *hblA*, *hblC*, and *nheC* were detected in 74.6, 61.9, and 49.2% of the isolates, respectively. Furthermore, a total of 8 (4.4%) and 22 (12%) out of 181 *B. cereus* isolates harboured all three genes encoding the enterotoxic NHE (ABC) and HBL (ACD) complexes, respectively. This finding is in agreement with other previous studies, which showed that most of the *B. cereus* strains isolated from food carried all three genes of the *nhe* operon [45,46,47,48,49]. Additionally, Sánchez-Chica et al. [49] detected 30 *B. cereus* isolates (57%) carrying the HBL (ACD) complex. Similar HBL (ACD) prevalences have been reported previously, showing results of about 50–70% [45,46,50,51]. Compared to our study, the prevalence of both NHE and HBL complete complexes reported in the aforementioned studies was much higher, highlighting the variability in the occurrence of these complexes, which can be ascribed to different food products and geographical distribution. By analyzing the hemolytic (HBLs) and nonhemolytic (NHEs) enterotoxins genes individually, our results showed higher prevalences than the corresponding complete complex, and at least one gene from the *nhe* and *hbl* operon was found, as in the previous studies by Ceuppens et al. [4] and Tewari et al. [52]. Guo et al. [46] reported higher prevalence rates of the *nheA*, *nheB*, and *nheC* genes in strains isolated from quick-frozen food (both processed and non-processed samples), at 100.0%, 100.0%, and 88.8%, respectively, compared to our present study’s findings (85.1%, 81.8%, and 49.2%). Additionally, *hblA*, *hblC*, and *hblD* were found in 65.2%, 86.2%, and 75.0% of the strains [46], contrasting with the rates in our study, which showed 74.6%, 61.9%, and 85.6%, respectively. Various authors have consistently reported that genes from the HBL complex are less prevalent than those from the NHE complex [41,52,53]. On the other extreme, a very low prevalence of the cytK gene was observed (1.1%; 2/181). This is a lower prevalence rate when comparing with other studies on different food products [38,46,52,54,55]. Studies in Europe and around the world have shown that the distribution of the enterotoxin gene *cytK* is around 40–70% in fresh vegetables [42], quick-frozen food [46], and flour products (including, wheat flour, spelt flour, rye flour, and flour mixes) [56]; however, this prevalence can be higher in other food products including cereals, ready-to-eat meals, spices, dairy products, and starches [57,58].

Regarding the gene encoding emetic toxin (*ces*), it was detected in 3.3% (6/181) of the isolates (Table 5). Despite being present in a low percentage of isolates, it should be highlighted that the *ces* gene was detected in isolates recovered from blackberry and strawberry final products (50%; 3/6; Figure 2). These results seem to be in agreement with previous studies on the virulence gene profiles of *B. cereus*, showing that the prevalence of strains producing emetic toxin is rather low [41,42,46]. The presence of the emetic toxin is usually related to starchy food products, including pasta, potatoes, and rice, but can also be found in soups, sauces, and other cooked food items [59,60,61].

Figure 2 represents the distribution of toxin profiles in berries samples at different stages of processing. The toxin profiles were mostly present in raw materials and intermediate products. In the case of blackberries and strawberries, two profiles were also detected in isolates recovered from final product samples. The enterotoxin gene *cytK* (toxin profiles II and XIX) was detected only in isolates from raw blueberry and raspberry samples, while the *ces* gene (toxin profile I) was found in isolates from the final products of strawberry and blackberry isolates. Noticeably, the eight toxin genes investigated were not detected in isolates from the final products of raspberry and blueberry samples.

Among all the fruits studied, blueberries showed the highest number of toxin profiles in raw materials followed by blackberries and raspberries. Regarding intermediate product samples, raspberries presented seven different toxin profiles, while a lower variation was observed in the other fruits.

### 3.2. Cytotoxic Activity of B. cereus Isolates

The evaluation of the cytotoxic activity on Caco-2 cells was performed only for the isolates carrying the *cytK* gene (two isolates, toxin profiles II and XIX; Table 4). After growing for 18 h and 5 days at 30 °C, the culture supernatants of these isolates showed cytotoxic activity on Caco-2 cells, with a percentage higher than 60% and no differences (*p* > 0.05) between them were observed (Table 6). Additionally, this cytotoxic activity was similar (*p* > 0.05) to the one observed by the strain used as a positive control, *B. cereus* subsp. cytotoxicus NVH 391-98. Similarly, Gdoura-Ben Amor et al. [41] reported 70.7% cytotoxic activity for the majority of the isolates tested after 18 h of incubation. On the other hand, the authors observed a decrease in cytotoxic activity after 5 days, probably due to the degradation of toxins and the death of the cells. Jan et al. [62] observed a cytotoxic activity of 55.1 and 67.9% after 18 h and 5 days at 30 °C, respectively. Although our results are representative of just two *B. cereus* isolates, this finding highlights the importance of controlling this pathogen from raw material to final products.

## 4. Conclusions

In conclusion, the results from this study showed the importance of considering *B. cereus*, a health hazard present in raw berries and their products due to the high prevalence of enterotoxin genes widespread among the isolates. All the isolates analysed in this study showed at least one of the toxin genes examined. Despite the fact that the emetic toxin gene was found in only 3.3% of the isolates, it was present in some berry final products, converting it to a target that should be monitored along with enterotoxin-producing *B. cereus* isolates.

## Figures and Tables

**Figure 1 foods-12-04021-f001:**
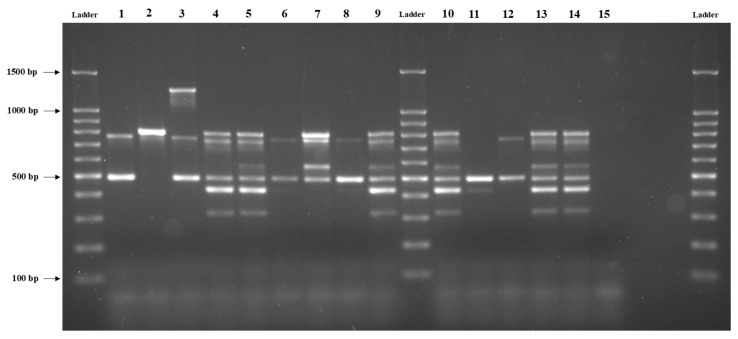
PCR products of strains with different toxigenic potential after Multiplex PCR. Lane 1, strain NVH 0075/95 (*nhe* reference); lane 2, strain NVH 391-98 (*cytK* reference); lane 3, strain DSMZ 4312 (cereulide and *nhe* reference); lane 4, strain DSMZ 4313 (*hbl/nhe* reference); lanes 5–9, 10–14 = *B. cereus* isolates and lane 15, sterile water (negative control).

**Figure 2 foods-12-04021-f002:**
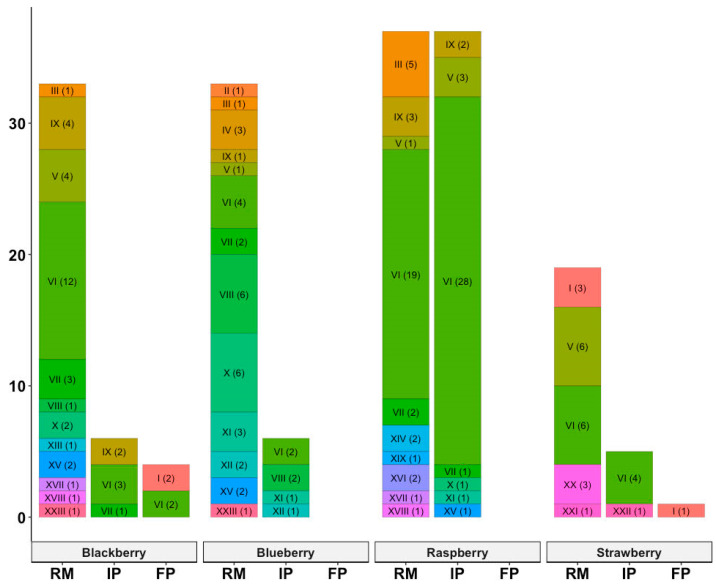
Total distribution of the different toxin gene profiles among the four types of berries at different stages of processing (RM—Raw Material; IP—Intermediate fruit product; FP—Final Product).

**Table 1 foods-12-04021-t001:** Number of *B. cereus* isolates recovered from berry fruit samples.

Strawberries			Blueberries			Raspberries			Blackberries		
RM	IP	FP	RM	IP	FP	RM	IP	FP	RM	IP	FP
19	5	1	33	6	0	37	37	0	33	6	4

RM: raw material; IP: intermediate product; FP: final product.

**Table 2 foods-12-04021-t002:** Sequences of PCR primers targeting various virulent factor genes in this study.

Target Gene	Primer	Primer Sequence (5′-3′)	Amplicon Size (bp)	Reference
*nheA*	nheA 344 S	TACGCTAAGGAGGGGCA	480	[26]
	nheA 843 A	GTTTTTATTGCTTCATCGGCT		
*nheB*	nheB 1500 S	CTATCAGCACTTATGGCAG	754	[26]
	nheB 2269 A	ACTCCTAGCGGTGTTCC		
*nheC*	nheC 2820 S	CGGTAGTGATTGCTGGG	564	[26]
	nheC 3401 A	CAGCATTCGTACTTGCCAA		
*hblA*	HBLA1	GTGCAGATGTTGATGCCGAT	301	[26]
	HBLA2	ATGCCACTGCGTGGACATAT		
*hblC*	L2A	AATGGTCATCGGAACTCTAT	731	[26]
	L2B	CTCGCTGTTCTGCTGTTAAT		
*hblD*	L1A	AATCAAGAGCTGTCACGAAT	411	[26]
	L1B	CACCAATTGACCATGCTAAT		
*cytK*	CK-F-1859	ACAGATATCGG(GT)CAAAATGC	809	[38]
	CK-R-2668	TCCAACCCAGTT(AT)(GC)CAGTTC		
*ces*	cesF1	GGTGACACATTATCATATAAGGTG	1271	[35]
	cesR2	GTAAGCGAACCTGTCTGTAACAACA		

**Table 3 foods-12-04021-t003:** Multiplex PCR conditions used.

Target Gene	Amplification Conditions		Expected Fragment Length bp
	Cycles	Conditions	
*hblA*, *hblC*, *hblD*, *nheA*, *nheB*, *nheC*, *cytK*, *ces*	1	Start cycle: 95 °C for 3 min	Between 300 and 1270
	35	Denaturation: 94 °C for 30 s	
		Annealing: 58 °C for 45 s	
		Extension: 72 °C for 1.50 min	
	1	Final extension: 72 °C for 5 min	
	∞	Storage: 4 °C	

**Table 4 foods-12-04021-t004:** Toxin gene profiles of *B. cereus* isolated from berry samples.

Toxin Profile Number	Virulence Genes	Number of Isolates (%)
I	*ces nheA nheB*	6 (3.3)
II	*cytK nheC nheA*	1 (0.6)
III	*hblD hblA*	7 (3.9)
IV	*nheA*	3 (1.7)
V	*nheB hblC nheA hblD hblA*	15 (8.3)
VI	*nheB hblC nheC nheA hblD hblA*	80 (44.2)
VII	*nheB hblD hblA*	9 (5.0)
VIII	*nheB nheA*	9 (5.0)
IX	*nheB nheA hblD hblA*	12 (6.6)
X	*nheA hblD*	9 (5.0)
XI	*nheB hblC nheC nheA*	5 (2.8)
XII	*nheB nheA hblD*	3 (1.7)
XIII	*hblC nheA hblD hblA*	1 (0.6)
XIV	*nheA hblD hblA*	2 (1.1)
XV	*hlbD hblC hblA*	5 (2.8)
XVI	*nheB hblC nheA hblD*	2 (1.1)
XVII	*hblD*	2 (1.1)
XVIII	*nheB hblD*	2 (1.1)
XIX	*cytK*	1 (0.6)
XX	*nheB nheC nheA hblD hblA*	3 (1.7)
XXI	*nheB hblC hblD hblA*	1 (0.6)
XXII	*nheB hblC nheA*	1 (0.6)
XXIII	*hblC nheA hblD*	2 (1.1)

**Table 5 foods-12-04021-t005:** Distribution of genes associated with pathogenesis in *B. cereus* isolated from different fruit matrices.

Virulence Genes	Number (%) of *B. cereus*				
	Total (n = 181)	Blackberries (n = 43)	Raspberries (n = 74)	Strawberries (n = 25)	Blueberries (n = 39)
** *ces* **	6 (3.3)	2 (4.6)	0	4 (16)	0
** *hblA* **	135 (74.6)	35 (81.4)	67 (90.5)	20 (80)	13 (33.3)
** *hblC* **	112 (61.9)	25 (58.1)	55 (74.3)	18 (72)	14 (35.9)
** *hblD* **	155 (85.6)	40 (93.0)	72 (97.3)	20 (80)	23 (59)
** *nheA* **	154 (85.1)	34 (79.1)	62 (83.8)	24 (96)	34 (87.2)
** *nheB* **	148 (81.8)	35 (81.4)	63 (85.1)	25 (100)	25 (64.1)
** *nheC* **	89 (49.2)	17 (39.5)	48 (64.9)	13 (52)	11 (28.2)
** *cytK* **	2 (1.1)	0	1 (1.3)	0	1 (2.6)

**Table 6 foods-12-04021-t006:** Cytotoxic activity of *B. cereus* isolates carrying the *cytK* gene on Caco-2 cells (%; mean ± standard deviation).

	Cytotoxic Activity (%) *	
Bacteria Isolates	18 h	5 Days
*Bacillus cereus* subsp. *cytotoxicus* NVH 391-98	64.4 ± 8.0	73.7 ± 12.0
BactoFlavor ScarLet	8.5 ± 0.5	1.7 ± 0.3
*B. cereus* isolate profile II	68.2 ± 11.2	66.4 ± 5.3
*B. cereus* isolate profile XIX	64.5 ± 9.1	64.0 ± 2.3

* No significant differences were found between all bacteria tested in all timepoints tested.

## Data Availability

The data used to support the findings of this study can be made available by the corresponding author upon request.

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
