# Peer review of "Characterization of the Toxigenic Potential of Bacillus cereus sensu lato Isolated from Raw Berries and Their Products"

_foods, 2023, doi:10.3390/foods12214021_

Round 1
Reviewer 1 Report
Comments and Suggestions for Authors
The topic of the manuscript (foods-2684631-peer-review-v1) is of interest because the fruit and vegetable consumption is constantly growing. The manuscript describes in general an interesting laboratory work, which could be useful for other researchers and/or industry working on the topic. In the study, the toxigenic potential of 181 isolates of Bacillus cereus were investigated by assessing the presence of enterotoxin genes and an emetic toxin cereulide synthetase gene. The cytotoxic activity on Caco-2 cells was also evaluated for the two isolates containing the gene cytK. The results found that cytotoxic activity on Caco-2 cells was also observed to be greater than 60% for isolates containing the cytK gene. Regarding results, some concepts should be carefully revised. There are many questions that have not been explained clearly.
1. Whether these two genes are affected by food processing?
2. Which substances are expressed by the two genes?
3. What are the contents of these substances?
4. Whether these substances can have toxic effects?
Author Response
R: Thank you for taking the time to review this manuscript and for your suggestions for improvement
The topic of the manuscript (foods-2684631-peer-review-v1) is of interest because the fruit and vegetable consumption is constantly growing. The manuscript describes in general an interesting laboratory work, which could be useful for other researchers and/or industry working on the topic. In the study, the toxigenic potential of 181 isolates of Bacillus cereus were investigated by assessing the presence of enterotoxin genes and an emetic toxin cereulide synthetase gene. The cytotoxic activity on Caco-2 cells was also evaluated for the two isolates containing the gene cytK. The results found that cytotoxic activity on Caco-2 cells was also observed to be greater than 60% for isolates containing the cytK gene. Regarding results, some concepts should be carefully revised. There are many questions that have not been explained clearly.
- Whether these two genes are affected by food processing?
R: New information added
- Which substances are expressed by the two genes?
R: I think the question is related to cytK gene in two isolates. Bacillus cereus gastrointestinal diseases are caused by a large number of virulence factors that are not yet fully understood. The level of expression of genes that play a role in its pathogenicity (e.g. genes encoding for toxins, such as hemolysins, enterotoxins, and emetic toxins, as well as enzymes involved in tissue degradation) determine the actual risk of food poisoning.
- What are the contents of these substances?
R: New information added
- Whether these substances can have toxic effects?
R: New information added
Reviewer 2 Report
Comments and Suggestions for Authors
The work is meaningful and the results are credible.
Here are some specific comments for improvement of the manuscript.
1. The introduction should provide more specific information.
2. Line 54 and 59, B. cereus is Bacillus cereus?Please clarify it.
3. Figure 2 is blurry, please revise it.
4. Line 210(Table 4) and 212(Table 5), the numbers in parentheses need to indicate their meaning.
5. Line 223-239, Discussion needs to be strengthened and literature should be cited.
6. Text formatting needs to be checked carefully. Such as Lines 236-240, there is a different Line Spacing.
7. Lines 259-264, The conclusion should be a separate part.
8. Lines 259-264, Some key data should be provided in the conclusion.
9. Some latest literature (after 2021) should be added.
10. One or more literature from this journal(Foods)should be cited.
Author Response
R: Thank you for taking the time to review this manuscript and for your suggestions for improvement
The work is meaningful and the results are credible.
Here are some specific comments for improvement of the manuscript.
- The introduction should provide more specific information.
R: As suggested, more information was added to the introduction.
- Line 54 and 59, B. cereus is Bacillus cereus?Please clarify it.
R: Yes. After its first genus name was abbreviated (exception beginning of a sentence)
- Figure 2 is blurry, please revise it.
R: The figure was pasted again.
- Line 210(Table 4) and 212(Table 5), the numbers in parentheses need to indicate their meaning.
R: The number in parentheses refers to the percentage. This is indicated in the table "(%)"
- Line 223-239, Discussion needs to be strengthened and literature should be cited.
R: Lines 223-239 are just a description of the results observed in this study.
- Text formatting needs to be checked carefully. Such as Lines 236-240, there is a different Line Spacing.
R: Thanks for spotting this typo
- Lines 259-264, The conclusion should be a separate part.
R: Modified as suggested
- Lines 259-264, Some key data should be provided in the conclusion.
R: Added as suggested
- Some latest literature (after 2021) should be added.
R: Added as suggested
- One or more literature from this journal(Foods)should be cited.
R: Added as suggested
Reviewer 3 Report
Comments and Suggestions for Authors
The manuscript by Olivera et al., describes the characterisation of toxin- related genes in 181 B. cereus strains isolated from berries. The manuscript is well written and the study is scientifically sound. The manuscript would be improved with some clarifications:
The authors previous manuscript described the B. cereus isolates as presumptive, please explain what additional testing was performed to confirm the isolates as B. cereus
There is no phylogenetic information on B. cereus and the findings of genomic studies of the B. cereus toxin distribution (e.g. Zhang et al., PLoS One. 2022; 17(5): e0262974 and Bohm et al., BMC Evol Biol 2015 Nov 10:15:246) need to be addressed in the paper.
Twenty three profiles are proposed in the paper, can the authors specifically comment whether all or some of their profiles are described in other studies. including the genome studies. The statement 'In general, the prevalence of the nhe and hbl operon genes observed in the isolates was similar (L194)' is vague.
L60 Correct '104 to105 cells'
Author Response
R: Thank you for taking the time to review this manuscript and for your suggestions for improvement
The manuscript by Olivera et al., describes the characterisation of toxin- related genes in 181 B. cereus strains isolated from berries. The manuscript is well written and the study is scientifically sound. The manuscript would be improved with some clarifications:
The authors previous manuscript described the B. cereus isolates as presumptive, please explain what additional testing was performed to confirm the isolates as B. cereus
R: No further tests were conducted. This information was added in the materials and methods section.
There is no phylogenetic information on B. cereus and the findings of genomic studies of the B. cereus toxin distribution (e.g. Zhang et al., PLoS One. 2022; 17(5): e0262974 and Bohm et al., BMC Evol Biol 2015 Nov 10:15:246) need to be addressed in the paper.
R: More information added to the introduction:
The Bacillus cereus group, also called B. cereus sensu lato (s.l.), is a group of genetically closely related species that includes pathogenic and non-pathogenic species [4]. As far as the authors know at the time of writing, the group comprises nine closely related bacterial species including, B. anthracis [5], B. cereus sensu stricto (s.s.), B. cytotoxicus [6], B. mycoides [7], B. pseudomycoides [8], B. thuringiensis, B. toyonensis [9], B. weihenstephanensis [7], and B. wiedmannii [10]. Their genetic similarity has been widely studied and is characterized by having significant implications for human health, agriculture and food processing [11]. The main species recognized as human pathogens are B. cereus s.s. most commonly as-sociated with food poisoning, B. anthracis, the causative agent of anthrax and B. cytotoxicus, a thermotolerant species also associated with food poisoning [12].
In a shift towards a strain-based approach to characterisation and risk assessment, a new taxonomic framework for B. cereus s.l. was proposed by Carroll et al. [13]. The aim was to provide a more accurate assessment of the pathogenic potential of B. cereus s.l. isolates while maintaining genomic species definitions. Within this updated framework, the previously known B. cereus s.l. species were reclassified into eight genomic species. In addition, three distinct biovars were created: Emeticus, Anthracis and Thuringiensis, based on the presence of genes associated with specific characteristics. More recentely, Zhang et al. [14] conducted a study that examined the species diversity of B. cereus strains and their phylogenetic relationships concerning virulence factors. The aim of the research was to systematically evaluate the distribution of virulence genes and analyze the structures and functions of virulence proteins. By assessing enterotoxicity through nheABC, hblACD, and cytK gene expression, the study classified the strains into three levels of toxicity. The results suggested that B. cereus strains have evolved into clusters characterized as non-toxic risk and toxic risk, further subdivided into medium-high-risk and medium-low-risk subclusters.
Twenty three profiles are proposed in the paper, can the authors specifically comment whether all or some of their profiles are described in other studies. including the genome studies. The statement 'In general, the prevalence of the nhe and hbl operon genes observed in the isolates was similar (L194)' is vague.
R: Modified to make it more informative
L60 Correct '104 to105 cells'
R: Thanks for spotting this typo
Round 2
Reviewer 2 Report
Comments and Suggestions for Authors
I give the green light to this paper.